# Inflammation as Prognostic Hallmark of Clinical Outcome in Patients with SARS-CoV-2 Infection

**DOI:** 10.3390/life13020322

**Published:** 2023-01-23

**Authors:** Diana Fuzio, Angelo Michele Inchingolo, Vitalba Ruggieri, Massimo Fasano, Maria Federico, Manuela Mandorino, Lavinia Dirienzo, Salvatore Scacco, Alessandro Rizzello, Maurizio Delvecchio, Massimiliano Parise, Roberto Rana, Nicola Faccilongo, Biagio Rapone, Francesco Inchingolo, Antonio Mancini, Maria Celeste Fatone, Antonio Gnoni, Gianna Dipalma, Giovanni Dirienzo

**Affiliations:** 1Clinical Pathology Unit, Murgia Hospital, Asl Bari, 70022 Altamura, Italy; 2Department of Interdisciplinary Medicine, School of Medicine, University of Bari “Aldo Moro”, 70124 Bari, Italy; 3Infectious Diseases Complex Unit, Murgia Hospital, Asl Bari, 70022 Altamura, Italy; 4Universitary Clinical Pathology Complex Unit, School of Medicine, University of Bari “Aldo Moro”, 70124 Bari, Italy; 5Department of Basic Medical Sciences, Neuroscience and Sense Organs, University of Bari “Aldo Moro”, 70124 Bari, Italy; 6Metabolic Disorders and Diabetes Unit, “Giovanni XXIII” Children’s Hospital, School of Medicine, University of Bari “Aldo Moro”, 70124 Bari, Italy; 7Department of Economics, University of Foggia, 71100 Foggia, Italy; 8PTA Trani, ASL BT, Internal Medicine Surgery, Viale Padre Pio, 76125 Trani, Italy

**Keywords:** biomarkers, COVID-19, C-reactive protein, interleukin-6, inflammation, in-hospital mortality, lactate dehydrogenase, lymphocytes, pneumonia, procalcitonin

## Abstract

Severe acute respiratory syndrome coronavirus 2 (SARS-CoV-2) is often characterized by a life-threatening interstitial pneumonia requiring hospitalization. The aim of this retrospective cohort study is to identify hallmarks of in-hospital mortality in patients affected by Coronavirus Disease 19 (COVID-19). A total of 150 patients admitted for COVID-19 from March to June 2021 to “F. Perinei” Murgia Hospital in Altamura, Italy, were divided into survivors (*n* = 100) and non-survivors groups (*n* = 50). Blood counts, inflammation-related biomarkers and lymphocyte subsets were analyzed into two groups in the first 24 h after admission and compared by Student’s t-test. A multivariable logistic analysis was performed to identify independent risk factors associated with in-hospital mortality. Total lymphocyte count and CD3^+^ and CD4^+^ CD8^+^ T lymphocyte subsets were significantly lower in non-survivors. Serum levels of interleukin-6 (IL-6), lactate dehydrogenase (LDH), C-reactive protein (CRP) and procalcitonin (PCT) were significantly higher in non-survivors. Age > 65 years and presence of comorbidities were identified as independent risk factors associated with in-hospital mortality, while IL-6 and LDH showed a borderline significance. According to our results, markers of inflammation and lymphocytopenia predict in-hospital mortality in COVID-19.

## 1. Introduction

From March 2019 to the present, the pandemic caused by a novel betacoronavirus, named severe acute respiratory syndrome coronavirus 2 (SARS-CoV-2), has quickly spread from the city of Wuhan (China) to the entire world, with a variable trend characterized by flattening periods of the pandemic curve and periods of re-emergency [1]. Up-to-date information from the World Health Organization (WHO) reports 663,248,631 confirmed cases of Coronavirus Disease 19 (COVID-19) and 6,709,387 deaths (https://covid19.who.int/, accessed on 19 January 2023). The clinical course of COVID-19 is characterized by markedly divergent clinical manifestations. Aside from asymptomatic or pauci-symptomatic cases, nearly 20% of patients had severe bilateral interstitial pneumonia, which was associated with a rapid deterioration of clinical condition and necessitated hospitalization [2]. In COVID-19, multiple organ dysfunction syndrome (MODS) is triggered by an immune-mediated systemic inflammation, which often leads to death [3]. This pro-inflammatory state is reflected in specific laboratory parameters, such as blood count and markers of systemic inflammation. In severe COVID-19, some altered parameters are common to septic states, such as C-reactive protein (CRP), procalcitonin (PCT), ferritin, D-dimers and fibrinogen, while others indicate, more specifically, a hyper-activation of the immune system and are often present in systemic autoimmune diseases, such as pro-inflammatory cytokine levels, e.g., interleukin-6 (IL-6) and interleukin-10 (IL-10), and reduction of both total lymphocytes and specific lymphocyte subsets [4,5,6].

The understanding of the clinical and biochemical effects of COVID-19 is constantly expanding. Several strategies for dealing with this global emergency have been developed, ranging from preventive measures, from individual protection devices, isolation of positive cases and vaccines, to adjuvant therapies and specific antiviral treatments [7,8,9]. Due to the extreme variety of the spectrum of clinical manifestations of COVID-19, it is advisable to tailor preventive and therapeutic choices according to the characteristics of the patients. For instance, it is worth noting that elder patients and patients with pre-existing pathological conditions and comorbidities are at a higher risk of developing a severe outcome of COVID-19 [10]. Thus, it is crucial for physicians to distinguish, as soon as possible, the patients who will suffer from severe illness and are at risk of death, especially in the hospital setting during pandemic waves.

The aim of this retrospective study is to detail the clinical and laboratoristic features of hospitalized COVID-19 patients in a center of Southern Italy, identifying significant differences between the groups of survivors and non-survivors, and independent factors of death, in an attempt to define threshold values indicative of patient outcome.

## 2. Materials and Methods

### 2.1. Study Design and Participants

A total of 150 patients hospitalized from March to June 2021 in the public hospital “F. Perinei” Murgia Hospital in Altamura, Italy, were enrolled; patients were divided into survivors (*n* = 100, group 1) and non-survivors (*n* = 50, group 2) according to the clinical outcome. This study was approved by the Ethics Committee of “Azienda Ospedaliero Universitaria Consorziale Policlinico”, Bari, Italy (0015987, 17 February 2022), and it was conducted in accordance with the Declaration of Helsinki for human studies.

### 2.2. COVID-19 RT-PCR Assay for Nasal and Pharyngeal Swab Specimens

Real-time reverse-transcriptase polymerase-chain reaction (RT-PCR) assay for nasal and pharyngeal swab specimens was employed to confirm COVID-19. Briefly, RNA extraction was performed using a NeoPlex COVID-19 Detection kit (GeneMatrix Inc., Temecula, CA, USA) on a KingFisher Extraction System (ThermoFisher Scientific, Waltham, MA, USA), according to the manufacturer’s instructions. Amplification conditions included reverse transcription at 50 °C for 30 min, denaturation at 95 °C for 15 min and 40 cycles of 95 °C for 15 s and 60 °C for 60 s for fluoresce detection. A cycle threshold value (Ct-value) ≤ 38 was defined as a positive test, following Centers of Disease Control and Prevention (CDC) recommendations.

### 2.3. Laboratory Medicine Analyses and Clinical Data Collection

Laboratory medicine analyses (e.g., blood routine, lymphocyte subsets and inflammation-related biomarkers) were performed on patients’ blood samples collected and analyzed in the first 24 h after admission to the Infective Diseases Department of “F. Perinei” Murgia Hospital. The total number of lymphocytes in peripheral blood was counted with an automated hematology analyzer (Pentra ABX, HORIBA, Kyoto, Japan). An Olympus AU680 (Beckman Coulter Brea, CA, USA) was used to collect LDH and CRP data. PCT was determined by Liaison (DiaSorin S.p.A., Saluggia, Italy). IL-6 and ferritin were measured by ADVIA Centaur XP Immunoassay System (SIEMENS Health, Erlangen, Germany, GmbH). D-Dimers and Fibrinogen values were measured by ACL TOP 500 (Werfen, Bedford, MA USA). Peripheral blood lymphocyte typing was performed in patients’ whole blood samples by cytofluorimetric analysis using AQUIOS CL Flow Cytometer (AQUIOS—Beckman Coulter CA, USA). Antibodies used for cell staining were TETRA-1 Panel (CD45, CD4, CD8 and CD3), and obtained data were analyzed using flow cytometry analysis software (Aquios system software, V2.2.0).

### 2.4. Statistical Analysis

Statistical analysis was performed by setting categorical variables as frequency rates and percentages, and continuous variables as means and 95% confidence intervals (95% CI) or median and interquartile range (IQR) values. The comparison of means for continuous variables that were normally distributed was performed with Student’s *t*-test. The Mann–Whitney U test was used for continuous variables there were not normally distributed. Proportions for categorical variables were compared using the χ2 test, while receiver operating characteristic (ROC) curve analysis was performed using the Wilson/Brown method (95% confidence interval and standard error). Multinomial binary logistic regression analysis results were reported as Odds Ratios (OR) with 95% CI. Statistical analyses were performed by MedCalc^®^ (Mariakerke, Belgium) and GraphPad Prism version 8.2 (GraphPad Software Inc., San Diego, CA, USA). Two-sided *p*-values lower than 0.05 were considered statistically significant.

## 3. Results

The median age of the patients was 70 years, showing a statistically significant difference between non-survivors and survivors (79 vs. 65 years, *p* < 0.001) (Table 1).

Male patients were significantly older than females (79 vs. 71 years), while the median period from the onset of the symptoms to hospital admission was the same in females and males (7 days). One hundred and twenty-six patients (84%) had comorbidities such as hypertension (55.3%), diabetes mellitus (24%), chronic cardiac disease (22.6%), malignancies (4.6%), obesity (35.3%), chronic pulmonary disease (8%), chronic kidney disease (7.3%) and chronic neurological disorders (16%) (Table 2).

All the patients with severe and moderate disease were given empirical antimicrobial treatment (cephalosporin, azithromycin and levofloxacin). Nineteen patients (12%) received antiviral therapy with remdesivir. In addition, all severe and moderate cases were administered corticosteroids (CTS) during hospitalization. Nine patients (6%) received hyperimmune plasma, and thirty-four patients (22%) required admission to the intensive care unit (ICU) (Table 3).

At baseline, 102 patients (68%) needed respiratory support by continuous positive airway pressure, 25 patients (16.7%) by a Venturi-type mask, 18 patients (12%) by a simple face mask. Compared with the reference range, significant differences in blood count, lymphocyte subsets and inflammatory-related biomarkers were observed between survivor and non-survivor groups (Table 4).

The median lymphocyte count was lower in the non-survivors group (*p* < 0.0001). When we tested different subsets of T cells, we found that even though both helper T cells (CD3^+^CD4^+^) and suppressor T cells (CD3^+^CD8^+^) in patients with COVID-19 were below reference range (CD3^+^CD4^+^: 500–1700 cells/µL, CD3^+^CD8^+^: 244–1100 cells/µL), the lowering of helper T cells was considerably pronounced in fatal cases (184 vs. 353 cells/uL; *p* < 0.0001). Suppressor T cells also showed a decreasing trend (83 vs. 172 cells/uL; *p* < 0.0001). Conversely, T-helper and T-suppressor ratio (CD4^+^/CD8^+^ ratio) remained in the normal range and showed no difference between the two subgroups. As shown in Figure 1, the area under the curve (AUC) derived from CD8^+^ T cells was as large as that derived from CD3^+^ cells or CD4^+^ cells (AUC CD8^+^ = 0.741 [0.655–0.827] vs. AUC CD3^+^ = 0.769 [0.690–0.848] or AUC CD4^+^ = 0.752 [0.670–0.833], *p* < 0.001).

Non-survivors had significantly higher serum levels of IL-6, LDH, CRP and PCT than survivors (Table 4). Conversely, except for fibrinogen, no significant differences were found in the levels of D-dimers and ferritin between the two groups. Figure 2 shows that the AUC of IL-6 was 0.735 [0.651–0.818] and LDH was 0.784 [0.703–0.864] (*p* < 0.001), and age and comorbidities had AUCs of 0.805 [0.736–0.873] and 0.709 [0.622–0.800], respectively.

By multivariable logistic regression analysis, two indicators were identified to be independent risk factors associated with in-hospital mortality: age > 65 years (OR = 1.14; 95% CI, 1.07–1.22, *p* = 0.0001) and number of comorbidities (OR = 1.84; 95% CI, 1.11–3.05; *p* = 0.0178). IL-6 levels > 20 pg/mL (OR = 1.03; 95% CI, 1.00–1.06) and LDH levels > 489 U/L (OR = 1.01; 95% CI, 1.00–1.01) showed a borderline 95% CI (Table 5).

## 4. Discussion

Infection by SARS-CoV-2 can cause sustained responses of pro-inflammatory cytokines and chemokines (namely, a “cytokine storm”), leading to a life-threatening immune-mediated MODS [3]. The identification of specific immunological and inflammatory profiles of patients, and their association with COVID-19 severity, is a challenge in order to promptly block systemic inflammation with targeted therapeutic interventions and, on the other hand, minimize unnecessary treatments, especially during pandemic waves.

In this study, we investigated the predictive values of markers of inflammation and lymphocytopenia in hospitalized severe COVID-19 patients, already assessed in other previous studies, with, in part, conflicting results. Our research showed that non-surviving hospitalized COVID-19 patients had significantly higher PCT, CRP, LDH and IL-6 levels, which are indicators of both in-hospital mortality and inflammation. Age and comorbidities—particularly hypertension, obesity, diabetes and chronic heart disease—were found to be independent risk factors linked to in-hospital mortality, while LDH and IL-6 showed a borderline significance.

PCT, the precursor of the hormone calcitonin, is an acute-phase glycoprotein produced by C-cells of thyroids and monocytes. PCT dramatically increases during bacterial and fungal infections while slightly increasing during viral infections, making it an important biomarker of sepsis. In our series of hospitalized COVID-19 patients, PCT levels correlated with disease severity. Data on the value of PCT as prognostic markers for COVID-19 are contradictory in the literature. When compared to moderate illness, PCT is four times higher in severe patients and eight times higher in critical patients, according to Hu R. et al. [11]. Likewise, several authors found that PCT levels are increased in patients with a fatal outcome of severe COVID-19 both at admission and during the course of hospitalization [10,12,13]. In addition, Sayah W. et al. assessed that PCT and the neutrophil-to-lymphocyte ratio are not influenced by the administration of CTS. For this reasons, they may constitute valid alternative markers to assess severe forms in patients already undergoing CTS [14]. The markedly increased levels of PCT in COVID-19 patients can be explained by several factors: a coexistent bacterial infection, prolonged invasive mechanical ventilation and the up-regulation of the signal transducer and activator of the transcription 3 (STAT3)-dependent pathway, which stimulates angiotensin-converting enzyme 2 (ACE2) and PCT production in monocytes [15,16,17]. Accordingly, other clinical trials refuted the negative prognostic value of PCT in severe COVID-19 [18,19].

CRP is a non-specific acute-phase glycoprotein produced by the liver in response to trauma, myocardial ischemia and infections. Bacterial infections usually determine a marked increase of CRP, while viral infections are associated with a mild increase in CRP levels. In our study, CRP was significantly higher in non-survivor COVID-19 patients compared with survivor groups. CRP proved to be one of the earliest negative prognostic markers in COVID-19 because its levels increased before the appearance of radiologic findings at chest computer tomography (CT) [20]. Furthermore, CRP levels increased both at the beginning and during the progression of COVID-19 disease, and correlated with severity and mortality [20,21]. Several authors tried to improve the predictive value of CRP by relating it to other parameters, such as the CRP/albumin ratio and the CRP/lymphocytes ratio [22,23]. In particular, CRP is associated with in-hospital mortality due to venous thromboembolism and acute kidney injury [24], and a value of CRP equal to or higher than 40 mg/L is considered life-threatening in COVID-19-hospitalized patients [25].

LDH is an enzyme of the oxidoreductase class produced by different cells that is released into the blood as a result of cell damage or high turnover, such as in cancer, trauma, inflammation or infection. Other authors have assessed that LDH is correlated with poor prognosis in hospitalized COVID-19 patients, also indicating lung and other tissue injuries [26,27]. Thus, COVID-19 may lead to inadequate tissue perfusion and MODS, causing LDH elevation [28]. Thus, high values of LDH could represent a valid biomarker of mortality due to widespread infection in COVID-19.

In our study, we also showed that lymphocyte counts in COVID-19 patients and the CD3^+^ and CD4^+^CD8^+^ subsets were significantly lower, especially in the non-survivor group. Lymphocytopenia indicates a dysregulation of the immune system and has been observed in COVID-19 patients showing different spectrums of clinical disease [5,24]. Consisting with the literature data, we found a significant decrease of total peripheral lymphocyte counts and T cell main subsets (CD3^+^ and CD4^+^CD8^+^) in both the survivor and non-survivor groups, even if these parameters were significantly lower in the latter group. The etiopathogenesis of lymphocytopenia in COVID-19 patients has been related to different causes. ACE2, identified as the main cell entry receptor for SARS-CoV-2, is low-expressed by lymphocytes, and the viral genome is rarely detectable in peripheral blood of infected patients [29]. Thus, it is reasonable to speculate that the decrease of peripheral lymphocytes is not ascribable to the direct damage of SARS-CoV-2 on lymphocytes, but rather to an exhaustion of them in terms both of number and function due to the persistent exposure to viral antigens [30,31]. An alternative explanation is that the decrease of peripheral lymphocytes is a result of activation-induced apoptosis or aggressive migration from peripheral blood to the lungs, where robust viral replication occurs [32]. Lymphocytopenia in COVID-19 may be related to hyper-activation of STING (stimulator of interferon genes) due to DNA damage following acute distress respiratory syndrome (ARDS). STING is able to activate the NF–κB pathway and also to determine a progressive CD4^+^CD8^+^ T lymphocytopenia, similar to what occurs in STING-associated vasculopathy with onset in infancy (SAVI) syndromes [33].

Several studies detected a significant reduction in total lymphocytes count and in CD3^+^ and CD4^+^CD8^+^ T-cell subsets, both at the early stages and in severe forms of COVID-19-associated disease in deceased hospitalized patients compared to survivors [34,35,36,37]. Interestingly, a Brazilian study found that reduction of T-cell subtypes is a prognostic factor not only of death but also of need for intubation [38]. In addition to lymphocyte subsets, a high neutrophil-to-lymphocyte ratio has been considered a sensible parameter of negative outcome in COVID-19 [38,39].

The reduction of lymphocytes in COVID-19 also affects B cells and NK (natural killer) cells, as suggested by other studies, highlighting a significant reduction of CD19^+^ and NK cell count in hospitalized patients, which correlated with progression of disease and death [34,40].

Cytokines have been thought to play an important role in immunity and immunopathology during virus infections. “Cytokines storm” is a phenomenon of excessive inflammatory reaction in which cytokines are rapidly produced in large amounts in response to microbial infection, as well as to therapies, pathogens, cancers, autoimmune conditions and monogenic disorders [41]. This phenomenon has been considered an important contributor to ARDS and MODS in COVID-19 patients [3]. It has been reported that the levels of IL-6, interleukin-2 (IL-2), interleukin-7 (IL-7), IL-10, tumor necrosis factor-alpha (TNF-α), granulocyte colony-stimulating factor (G-CSF), interferon gamma induced protein (IP-10), monocyte chemoattractant protein-1 (MCP-1) and macrophage inflammatory protein-1 alpha (MIP-1α) were significantly higher in COVID-19 patients [42,43,44,45,46].

IL-6 is a protein produced by various types of cells, particularly T lymphocytes, macrophages, mature adipocytes and myocytes, in response to tissue damage from trauma, infection or inflammation, exerting both pro-inflammatory and anti-inflammatory effects [47].

In several studies, IL-6 has been proven to be an independent factor predictive of in-hospital mortality, and its levels are not influenced by the administration of CTS [14,42,43]. In particular, high levels of IL-6 are indicative of lung involvement, acute kidney injury, brain damage, cardiovascular events, and, finally, intestinal permeability, which allows viruses to become widespread in the general circulation [48,49,50,51,52]. The value of IL-6 has been included in a score system that predicts the need for non-invasive ventilation by combining systemic inflammatory biomarkers and a chest CT severity score [53].

Besides IL-6 and LDH, other laboratory parameters are cited in the literature as independent risk factors of death in COVID-19, such as neutrophil count, platelet count, CRP, D-dimers, troponin-I and low total testosterone [54,55,56,57].

We found that age and comorbidities—primarily hypertension and obesity, followed by diabetes, chronic cardiac disease, chronic neurological disorders, chronic pulmonary disease, chronic kidney disease and malignancies—are independent risk factors of COVID-19 mortality [25,50]. Consistently, other studies reported that patients older than 65 suffering from comorbidities experienced more severe symptoms, MODS and death [27,58]. Hypertension and obesity are the most frequent comorbidities in patients with severe or fatal COVID-19, and the main cause of death after ARDS is a cardiovascular acute event, such as myocardial dysfunction, arrhythmia or shock [10,12,13,26,59,60]. Instead, according to other authors, COVID-19-deceased patients presented, at admission, more frequently with chronic kidney disease and neurological diseases [13]. Obesity, diabetes and chronic kidney disease are also independent risk factors for intubation, as well as death [61]. Other pre-existing pathological conditions, such as anemia, hypotension, dyslipidemia, hyperglycemia and use of CTS, have been reported as independent risk factors of severe COVID-19 [26,53,62,63].

## 5. Conclusions

SARS-CoV-2 induces serious infectious diseases and becomes a continuous threat to human health. A rapid and well-coordinated immune response is the first line of defense against viral infections. However, when the immune response is dysregulated, it will result in excessive inflammation, even causing death. The higher expression of proinflammatory cytokines in COVID-19 patients, especially in severe cases, with the consumption of CD4^+^CD8^+^ T cells might result in aggravated inflammatory responses, the production of cytokine storms and clinical conditions worsening.

Our findings demonstrated that lymphocyte counts and CD3^+^ and CD4^+^CD8^+^ subsets were significantly lower in COVID-19 patients, particularly in the non-survivor group. IL-6, LDH, CRP and PCT levels were significantly higher in non-survivor hospitalized patients affected by COVID-19, representing not only markers of inflammation but also in-hospital mortality. Age and comorbidities, especially hypertension, obesity, diabetes and chronic heart disease, were identified as independent risk factors associated with in-hospital mortality. Elevated LDH and IL-6 levels may be substantial risk factors for in-hospital mortality even though they did not achieve significance as independent risk variables.

Even though our findings resulted from a monocentric study and the time frame for patient enrollment was limited, they highlighted the pivotal role of IL-6, LDH, CRP and PCT in predicting mortality for hospitalized COVID-19 patients.

Nevertheless, the results of this study should be confirmed in further larger controlled trials, while also developing scoring systems to assess the severity of COVID-19 disease, correlating every stage to a specific risk of mortality. On the other hand, it appears critical to establish prognostic survival factors for COVID-19 patients in order to screen the most vulnerable patients, organize targeted therapeutic interventions and reduce healthcare costs and waste.

## Figures and Tables

**Figure 1 life-13-00322-f001:**
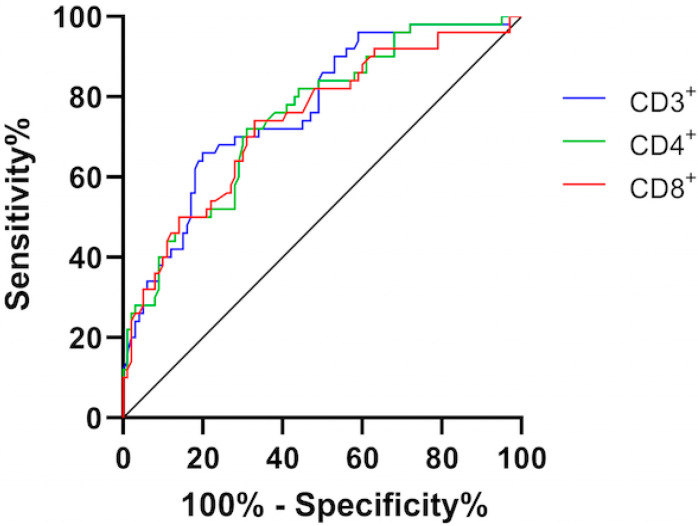
ROC curve of T lymphocyte subsets.

**Figure 2 life-13-00322-f002:**
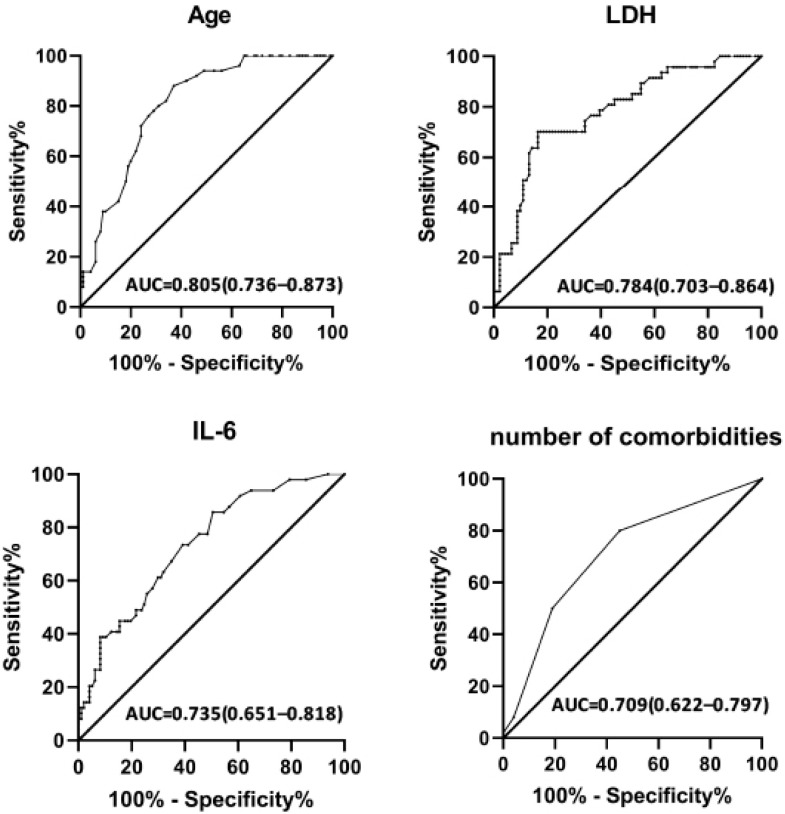
ROC curves of age, IL-6, LDH and number of comorbidities [AUC: area under the ROC curve].

**Table 1 life-13-00322-t001:** Baseline characteristics of 150 enrolled patients with COVID-19. Values are reported as median (IQR) or number of patients (rate%), as appropriate.

Characteristic	Patients (*n* = 150)	Survivors (*n* = 100)	Non-Survivors (*n* = 50)	*p*-Values
Age (years)	70 (62–80)	65 (57–73)	79 (73–85)	<0.0001
Males	79 (52.7%)	64 (55.25–69.75)	78 (69–83)	<0.0001
Females	71 (47.3%)	68 (57.25–77.5)	83 (74–85)	<0.0001
Days from clinical onset	7 (7–10)	7 (3–20)	7 (2–30)	0.994
Males	7 (7–10)	7 (7–10.75)	7 (7–10)	0.310
Females	7 (7–10)	7 (7–10)	7 (7–10)	0.392

**Table 2 life-13-00322-t002:** Comorbidities of 150 enrolled patients with COVID-19. Values are reported as number of patients (rate%).

Comorbidities	Patients (*n* = 150)
Hypertension	83 (55.3%)
Obesity	53 (35.3%)
Diabetes	36 (24%)
Chronic cardiac disease	34 (22.6%)
Chronic neurological disorders	24 (16%)
Chronic pulmonary disease	12 (8%)
Chronic kidney disease	11 (7.3%)
Malignancies	7 (4.6%)
Autoimmune disorders	6 (4%)
HIV	0

**Table 3 life-13-00322-t003:** Medical treatment of 150 enrolled patients with COVID-19 pneumonia. Values are reported as median (IQR) or number of patients (rate%), as appropriate.

Treatments	Patients (*n* = 150)
Antiviral therapy	
Remdesivir, No (%)	19 (12)
Immune therapy	
Hyperimmune plasma, No (%)	9 (6)
CTS therapy	
Prednisone or equivalent > 1.5 mg/kg/day	148 (98)
ICU admission, No (%)	34 (22)
Length of hospital, days, median (range)	13 (1–87)

**Table 4 life-13-00322-t004:** Laboratory data of SARS-CoV-2 patients on admission in the survivors and non-survivors groups and survival correlation [CRP: C-reactive protein, Hb: haemoglobin, IL-6: interleukin-6, LDH: lactate dehydrogenase, PCT: procalcitonin; PLT: platelets, WBC: white blood cells].

Blood Count	Patients (*n* = 150)	Survivors (*n* = 100)	Non-Survivors (*n* = 50)	*p*-Values
WBC (×10^3^/µL)	8 (6–11)	7.8 (5.5–10)	8.2 (5.7–11)	0.3957
Lymphocytes (×10^3^/µL)	0.9 (0.6–1.3)	1.0 (0.70–1.4)	0.70 (0.50–0.90)	<0.0001
Monocytes (×10^3^/µL)	0.5 (0.3–0.6)	0.50 (0.30–0.60)	0.45 (0.30–0.60)	0.7255
Neutrophil (×10^3^/µL)	6.2 (1–23.1)	6.0 (3.8–8.4)	6.7 (4.9–9.9)	0.1263
CD3 (cells/µL)	443 (281–707)	556 (358–836)	304 (194–491)	<0.000 1
CD4 (cells/µL)	272 (174–445)	353 (207–548)	184 (113–272)	<0.0001
CD8 (cells/µL)	135 (77–228)	172 (97–265)	83 (49–139)	<0.0001
CD4/CD8 ratio	1.98 (1.4–3.2)	1.97 (1.53–2.98)	2.00 (1.15–3.98)	0.9539
Hb (g/dL)	13.4 (12–14)	14(12–15)	13 (11–14)	0.0535
PLT (×10^3^/µL)	236 (179–310)	254 (197–318)	200 (160–258)	0.0013
**Inflammation-Related Biomarkers**	**Patients** **(*n* = 150)**	**Survivors** **(*n* = 100)**	**Non-Survivors** **(*n* = 50)**	***p*-Values**
D-Dimers (ng/mL)	1118 (696–2160)	1041 (652–1692)	1490 (997–3604)	0.0114
Ferritin (ng/mL)	738 (407–1293)	655 (349–1013)	1111 (618–1561)	0.0018
Fibrinogen (g/L)	533 (418–643)	523 (406–633)	544 (453–678)	0.4193
IL-6 (pg/mL)	13 (2–28)	9 (3–19)	20 (11–57)	<0.0001
LDH (U/L)	357 (272–464)	318 (257–398)	489 (359–557)	<0.0001
CRP (mg/L)	67 (26–124)	51 (16–80)	128 (68–170)	<0.0001
PCT (ng/mL)	0.08 (0.03–0.23)	0.05 (0.02–0.12)	0.21 (0.10–0.51)	<0.0001

**Table 5 life-13-00322-t005:** Variables independently associated with survival (odds ratios and 95% confidence intervals).

Variables	Odds Ratio	95% CI	*p*-Value
Age	1.14	1.07 to 1.22	0.0001
Number of comorbidities	1.84	1.11 to 3.05	0.0178
Variables not included in the model			
D-Dimers, Ferritin, Fibrinogen, CRP, PCT, IL-6, LDH			

## Data Availability

Not applicable.

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
