# Peer review of "Inflammation as Prognostic Hallmark of Clinical Outcome in Patients with SARS-CoV-2 Infection"

_life, 2023, doi:10.3390/life13020322_

Round 1

Reviewer 1 Report

Dear Authors, the article is well crafted and it is innovative and of a certain significance in its field. In my opinion, minor changes are required prior to pubblication. 

1) Please divide M&M sections in paragraph in order to be clearer (e.g.:Study population, Molecular analysis, Statistical analysis, etc...);

2) Please check English spelling/grammar in the whole manuscript.

Author Response

Point 1. Please divide M&M sections in paragraph in order to be clearer (e.g.:Study population, Molecular analysis, Statistical analysis, etc...)

Response 1. Thank you for your suggestion. We divided the M&M sections into paragraphs.

Point 2. Please check English spelling/grammar in the whole manuscript

Response 2. The entire manuscript was revised, checking for grammar and spelling errors. Please find an updated version of our manuscript attached. 

Reviewer 2 Report

The analysis seems valid and the authors acknowledge its limitations.

My criticisms are the following:

1) The Ethics committee approved the study in February 2022.  However, it appears the data was collected (patients enrolled, samples collected and tested) in March through June 2021.  I can understand this timeline if all of the data was gathered as part of routine clinical care, and the Ethics committee approved a completely retrospective study.  However, some of the data presented does not seem like it would routinely be collected on admission (for example, CD3 counts, or fibrinogen).  This should be clarified.

2) It is not clear which instruments performed which tests (lines 102-105)

3) As described in the "Discussion" section, many of the findings in this work have been shown previously.  The originality is limited.

4) The findings are likely not applicable to COVID-19 patients in the future.  The virus is mutating, new vaccines are being produced, and new therapies are being implemented.  

5) The bulk of the discussion is essentially a review article of inflammatory biomarkers.

6)  There is a sloppiness to the writing, independent of the grammatical problems.  For example, Procalcitonin is abbreviated PCT, but is written as "CPT" in table 4, and as "PTC" on line 192.  There are other examples, but I don't want to list them all.  

Author Response

Point 1. The Ethics committee approved the study in February 2022.  However, it appears the data was collected (patients enrolled, samples collected and tested) in March through June 2021.  I can understand this timeline if all of the data was gathered as part of routine clinical care, and the Ethics committee approved a completely retrospective study.  However, some of the data presented does not seem like it would routinely be collected on admission (for example, CD3 counts, or fibrinogen).  This should be clarified.

Response 1. Thank you for your comments. Study  approved by the Ethics Committee of "Azienda Ospedaliero Universitaria Consorziale Policlinico", Bari, Italy (0015987, 17 February 2022) was an observational, descriptive, non-interventional, retrospective and single-center, non-profit study related to the quantitative determination of changes in T lymphocyte subpopulations and changes in markers of the inflammatory response in COVID-19 patients. All the parameters presented, including CD3 counts and fibrinogen, were determined on patients admission in keeping with local protocol for COVID-19 patients, and laboratory data were retrospectively collected from medical records.

Point 2. It is not clear which instruments performed which tests (lines 102-105)

Response 2. We clarified this point, revising the text according to your suggestion.

Point 3. As described in the "Discussion" section, many of the findings in this work have been shown previously.  The originality is limited.

Response 3. Previous research has extensively investigated the predictive value of inflammation and lymphocytopenia markers in hospitalized severe COVID-19 patients. Nonetheless, the findings were, to some extent, contradictory. Our study added some new information to the existing data, partially validating it.

Point 4. The findings are likely not applicable to COVID-19 patients in the future.  The virus is mutating, new vaccines are being produced, and new therapies are being implemented.  

Response 4. We share your concern about the applicability of the current data in the future, considering the emergence of new viral variants. However, studies like ours have been indispensable in defining the immune response to COVID-19 and preventing the harmful effects of the disease, especially in the most vulnerable patients and in not yet vaccinated patients.

Point 5. The bulk of the discussion is essentially a review article of inflammatory biomarkers.

Response 5. We added some remarks about the findings of our study to the discussion section after considering your idea.

Point 6. There is a sloppiness to the writing, independent of the grammatical problems.  For example, Procalcitonin is abbreviated PCT, but is written as "CPT" in table 4, and as "PTC" on line 192.  There are other examples, but I don't want to list them all.  

Response 6. Thanks for your suggestion. We revised the entire text, making corrections where necessary.

Author Response

Dear reviewer, we appreciated your comments and modified the manuscript accordingly. We extensively revised the English, correcting grammar and spelling errors you reported. We also revisited manuscript accordingly to points you listed.

As concerns your question about multivariate logistic analysis, variables included were:

  • Age
  • Albumin
  • ALT
  • CD3_
  • CD4_
  • CD4_CD8_ratio
  • CD8_
  • D_Dimers
  • Days_from_onset
  • eGFR
  • Ferritin
  • Fibrinogen
  • Hgb
  • IL_6
  • LDH
  • Lenght_in_hospital
  • Lymphocytes__n_
  • Monocytes__n_
  • Neutrophil__n_
  • number_of_comorbidities
  • PCR
  • PLT
  • Procalcitonin
  • WBC

Regarding your inquiry about OR analysis, survival was taken into account when calculating the results. Age, LDH, IL-6 and number of comorbidities were included in the OR analysis.

In relation to your query about comorbidities, their number was included as one variable. For this reason, we could not infer which of the comorbidities is the most important in patients with a fatal outcome.

As concerns your question about multivariate logistic regression (line 175), and, in particular about the methodology of assessment the Cutoff values, our results and table 5 could be debatable indeed. A multinomial binary logistic regression analysis was run, this point has been clarified. In this version, we considered only variables with a 95% CI above 1. Variables with a 95% CI equal to 1 were eliminated.

In relation to your  comment : “In this study, LDH has proved to be an independent risk factor of in-hospital mortality in COVID-19 patients” and this is not necessarily true since according to table 5, the 95% CI for OR includes 1, meaning that it cannot be seen as an independent risk factor for mortality prediction", the sentence was cleared and paper modified accordingly.

Regarding your comment:  … "the authors must discuss the findings regarding IL 6 since in the present study also for IL6 the declared OR was 1.03 with 95% CI (1.0-1.06), thus it cannot be said that IL 6 is an independent significant predictive marker (Line 273 and beyond)", paper was modified in keeping with your suggestion.

Round 2

Reviewer 3 Report

The article has been improved. However, I think that in table 5 the Variables are independently associated with in-hospital mortality rather than with survival (as stated). 

Also since the IL-6 and LDH are clearly higher in non-survivors compared to survivors, the OR for IL-6 and LDH could be mentioned as risk factors, having a border significance, which in other conditions with an increased number of patients included, the results would be more valuable. Also, this marginal significance for IL-6 and LDH should be also mentioned within the abstract.

Author Response

Dear reviewer, we appreciated your suggestion. 

As you pointed out, and as is also indicated in the abstract section, the variables listed in table 5 are independently associated with in-hospital mortality. 

We also modified abstract, discussion and conclusion sections according your comments (changes were highlighted in yellow).